# Molecular Characterization of Coxsackievirus A24v from Feces and Conjunctiva Reveals Epidemiological Links

**DOI:** 10.3390/microorganisms9030531

**Published:** 2021-03-05

**Authors:** Magilé C. Fonseca, Mario Pupo-Meriño, Luis A. García-González, Mayra Muné, Sonia Resik, Heléne Norder, Luis Sarmiento

**Affiliations:** 1Virology Department, Center for Research Diagnosis, and Reference, Institute of Tropical Medicine “Pedro Kourí”, Havana 11400, Cuba; mayra@ipk.sld.cu (M.M.); sresik@ipk.sld.cu (S.R.); 2Departamento de Bioinformática, Universidad de las Ciencias Informáticas (UCI), Habana 19370, Cuba; mpupom@uci.cu; 3Centro de Estudios de Matemática Computacional, Universidad de las Ciencias Informáticas (UCI), Habana 19370, Cuba; lgarciaag89@gmail.com; 4Departamento de Ciencias de la Computación, Centro de Investigación Científica y de Educación Superior de Ensenada, 22860 Ensenada, Mexico; 5Department of Infectious Diseases/Virology, Institute of Biomedicine, Sahlgrenska Academy, University of Gothenburg, 40530 Gothenburg, Sweden; helene.norder@gu.se; 6Department of Clinical Microbiology, Region Västra Götaland, Sahlgrenska University Hospital, 41345 Gothenburg, Sweden; 7Immunovirology Unit, Department of Clinical Sciences, Skåne University Hospital, Lund University, 22185 Malmo, Sweden

**Keywords:** coxsackievirus A24v, enterovirus, acute hemorrhagic conjunctivitis, conjunctival swabs, feces

## Abstract

Coxsackievirus A24 variant (CVA24v), the main causative agent of acute hemorrhagic conjunctivitis (AHC), can be isolated from both the eyes and lower alimentary tract. However, the molecular features of CVA24v in feces is not well-documented. In this study, we compared the VP1 and 3C sequences of CVA24v strains isolated from feces during AHC epidemics in Cuba in 1997, 2003, and 2008–2009 with those obtained from conjunctival swabs during the same epidemic period. The sequence analyses of the 3C and VP1 region of stool isolates from the three epidemics showed a high degree of nucleotide identity (ranging from 97.3–100%) to the corresponding conjunctival isolates. The phylogenetic analysis showed that fecal CVA24v isolates from the 1997 and 2003 Cuban outbreaks formed a clade with CVA24v strains isolated from conjunctival swabs in Cuba and other countries during the same period. There were three amino acid changes (3C region) and one amino acid change (VP1 region) in seven CVA24v strains isolated sequentially over 20 days from fecal samples of one patient, suggesting viral replication in the intestine. Despite these substitutions, the virus from the conjunctival swab and fecal samples were genetically very similar. Therefore, fecal samples should be considered as a reliable alternative sample type for the routine molecular diagnosis and molecular epidemiology of CVA24v, also during outbreaks of AHC.

## 1. Introduction

Coxsackievirus A24 variant (CVA24v) is a major etiological agent of acute hemorrhagic conjunctivitis (AHC), which is an epidemic form of a highly contagious ocular disease [1,2]. CVA24v was first isolated in 1970 during an epidemic of AHC in Singapore and classified as a member of the genus Enterovirus in the family *Picornaviridae*. As such, CVA24v shares features of enteric pathogenesis that are similar to those found in most of the human enteroviruses [3,4,5]. Thus, while CVA24v is transmitted primarily by hand-to-eye-to-hand contact, it is believed that CVA24v could be transmitted via the fecal–oral routes, as with other enteroviruses. This alternative mode of transmission is thought to play a role in the rapid and widespread transmission of AHC. In fact, the dual tropism of CVA24v for eyes and airways correlates with the use of sialic acid as a cellular receptor, which is also a feature of the other two viruses with pandemic potential: enterovirus D70 and influenza A viruses [6,7,8].

Although fecal shedding and isolation of CVA24v from throat swabs have been consistently demonstrated in multiple studies [3,5,9,10], little information is currently available regarding the molecular characteristics of CVA24v strains recovered from throat swabs and stool samples. We previously isolated CVA24v in the throat, nasal swab, and fecal samples collected from patients with the clinical diagnosis of AHC during the Cuban epidemics of 1997, 2003, and 2008–2009 [11,12]. Remarkably, the recovery levels of CVA24v from fecal samples during the 2008–2009 epidemics were consistently higher than those from conjunctival swabs, supporting the view that feces may represent a good source for viral isolation and fecal–oral transmission of CVA24v [12].

In a previous study, we determined the phylodynamics and phylogenetic relationships between epidemic strains of CVA24v circulating in Cuba between 1986–2009 and CVA24v strains isolated from other parts of the world during the same periods [13]. Although the study addressed one of the puzzling questions of when and where the CVA24v epidemics in Cuba originated, the molecular features of CVA24v in feces have not yet been studied. In the current study, the sequences of VP1 and 3C genomic regions were compared in CVA24v strains isolated from feces and conjunctival swabs during three AHC outbreaks in 1997, 2003, and 2008–2009, with the aim to investigate the degree of molecular similarities between CVA24v isolates from these two sample types.

## 2. Materials and Methods

### 2.1. Samples

A total of 34 CVA24v strains isolated from feces from patients with AHC during the Cuban outbreaks of 1997, 2003, and 2008–2009 were included in the study (Appendix A). CVA24v from stool samples were obtained by isolation on Hep2 (HeLa derivative, ECACC 86030501). Sixteen stool isolates were obtained from nine patients with AHC during an outbreak in 1997. Seven of these isolates were derived from single samples from seven patients. Two isolates were from one patient, from whom CVA24v was also isolated in nasal and pharyngeal swabs, and seven isolates were derived from one patient with sequential fecal sampling. From an AHC outbreak in 2003, 16 stool isolates of CVA24v were available from the same number of patients. CVA24v isolated from conjunctival swabs from four of these patients was also available. Two stool isolates from the 2008–2009 outbreaks previously described by Fonseca et al. were also included in the study [12].

### 2.2. PCR Amplification and Partial Sequencing of VP1 and 3C Regions

Viral RNA was isolated from 1997 and 2003 CVA24v strains according to the *TRIzol* manufacturer’s protocol (Life Technologies, Gibco BRL., Grand Island, NY, USA). The QIAmp viral RNA Mini kit (QIAGEN GmbH, Hilden, Germany) was used to extract RNA from the 2008–2009 CVA24v strains, as recommended by the manufacturer. The partial VP1-coding region of the genome was amplified in a reverse transcription-PCR assay performed with the published primers 222 (5′-CICCIGGIGGIAYRWACAT-3′) and 224 (5′-GCIATGYTIGGIACICAYRT-3′) [14]. PCR amplification and sequencing primers D1 (5′-TACAAACTGTTTGCTGGGCA-3′) and U2 (5′-TTCTTTTGATGGTCTCAT-3′) were used for the analysis of the 3C protease region [15]. Sequencing was performed with the ABI BigDye^®^ Terminator v 3.1 Cycle Sequencing Kit (Applied Biosystems, Vantaa, Finland), using the same forward and reverse primers used for the PCR amplification. The nucleotide sequences of the CVA24v strains isolated in 1997, 2003, and 2008–2009 were obtained using the ABI PRISM 3100 Genetic Analyser (Applied Biosystems) and CEQ TM 8800 Sequence Analysis System (Beckman Coulter, California, USA), respectively [12,13]. The CVA24v sequences from this study were submitted to GenBank under accession numbers KC286934–KC286938, KC286940–KC286941, KC286944–KC286950, KC286954–KC286955, KC286995, KC286997, KC286999–KC287003, KC287006, KC287016–KC287023, KC286974–KC286976, KC286978–KC286979, KC286982–KC286988, KC286992–KC286993, KC287039, KC287041, KC287043–KC287047, KC287050, KC287060–KC287067.

### 2.3. Identity Analysis

The obtained sequences were edited, and the percentages of similarity between the CVA24v isolated from stool and swabs were determined using the Bioedit v7.0.5.3 program [16].

### 2.4. Phylogenetic Analysis

#### 2.4.1. Dataset and Multiple Alignment

All phylogenetic analyses were carried out through the following workflow: alignment (multiple sequences alignment), quality control, evolutionary model selection, and testing phylogeny (modified from [17]), as previously described by Fonseca et al. [13]. Nucleotide sequence information from stool isolates of CVA24v was also compared with other VP1 and 3C sequences available in GenBank. Since a few VP1 sequences from representative CAV24v isolated in geographic areas severely affected by AHC outbreaks before the 2000s are available in GenBank, we included the VP1 sequences of CVA24v isolates from the feces of non-AHC patients (acute flaccid paralysis [AFP] cases, nonhuman primates, and river water) in our phylogenetic analysis. Duplicate sequences were eliminated using the ElimDupes tool from the Los Alamos HIV database (http://www.hiv.lanl.gov (accessed on 19 October 2015)). Multiple sequence alignments were carried out by MAFFT v7 with the L-INS-I algorithm [18].

#### 2.4.2. Quality Control

The quality of the sequence alignments of the VP1 and 3C coding regions resulted from the filtering of duplex sequences. The partial nucleotide sequences of both coding regions of Cuban and worldwide CVA24v strains were explored using the DAMBE v6.0 software [19] and standard statistical test, as described by Xia et al. [20]. Genetic distances were calculated with the general time-reversible (GTR) model at positions 1 + 2 + 3. The likelihood mapping method was used to assess the noise in the signal, as implemented in the Tree-puzzle v5.3.rc16 program [21]. Recombination events within CVA24v alignments and different strains of enterovirus C species were also tested via the RDP4 v4.36 program [22].

### 2.5. Phylogenetic Analyses

The phylogenetic analyses were as described previously [13]. The best-fit model nucleotide substitution models were estimated using the jModelTest v2.1.4 program according to the Bayesian Information Criterion (BIC) [23]. Bayesian Inference (BI) analysis, which is implemented into the Bayesian Evolutionary Analysis Sampling Tree (BEAST) v1.8.4 program [24], was used to infer the CVA24v phylogenetics. The substitution rates per site per year and the time of the most recent common ancestor (TMRCA) with a 95% highest posterior density (HPD) was analyzed by using the Bayesian Markov Monte Carlo Chain approach (MCMC). The clock model was selected by estimating the marginal (log) likelihood of each model using the path sampling (PS) method described by Baele et al. (2012) [25]. This simulation was carried out for models with a strict or relaxed molecular clock using an exponential uncorrelated (UCED) or a lognormal uncorrelated distribution (UCLD). These models were combined with the Bayesian Skyride Plot (BSP), a nonparametric coalescence model serving as an epidemiological model for the tree priors [26]. Bayesian MCMC analyses were run with a chain of 70 million generations for the 3C coding region and a chain of 100 million generations for the VP1 coding region. Convergence parameters were identified by Tracer v1.7.1 (http://tree.bio.ed.ac.uk/software/tracer/ (accessed on 1 May 2018)). with an effective sample size (ESS) greater than 200 (ESS > 200). In all cases, the initial 10% of the run was used as “burn-in”. The Maximum Clade Credibility (MCC) tree was calculated by TreeAnnotator v1.8.4 and then visualized with FigTree v1.4.4 (http://tree.bio.ed.ac.uk/software/figtree/ (accessed on 25 November 2018)).

## 3. Results

### 3.1. Analysis of Cuban CVA24v Sequences from Different Specimens

The sequences obtained from the 3C (*n* = 16) and VP1 region (*n* = 14) in 16 CVA24v stool isolates from the 1997 AHC epidemic (Appendix A) showed a high degree of nucleotide identity (ranging from 99.0–100%) with the 3C (*n* = 25) and VP1 (*n* = 22) sequences (Table 1) obtained from conjunctival swabs during the same epidemic period in Cuba (Appendix A). The failure to obtain VP1 sequences in 12.5% (two strains) of CVA24v may be due to VP1 primers’ reliance on conserved amino acid motifs specific to the *Enterovirus* genus [14]. The progressive degradation of viral RNA over multiple freeze–thaw cycles and prolonged storage may also result in a failure to obtain sequence CVA24v products.

The 3C and VP1 nucleotide sequences were 100% identical in CVA24v strains isolated from the stool, nasal, and pharyngeal swabs from one patient. Likewise, seven CVA24v strains that were sequentially isolated over 20 days from the feces of the same individual (Appendix A) shared 99.3–100% and 99.8–100% nucleotide identity in the 3C and VP1 coding regions, respectively. The sequences of these strains had five nucleotide changes in the 3C region, resulting in three amino acid changes (T120V, T120V, and H161L), and two nucleotide changes that resulted in one aminoacid change (T99A) in the VP1 region.

Similar to what was observed in the analysis of the 1997 AHC epidemic, the sequence analyses of the 3C (16 sequences) and VP1 regions (16 sequences) in 16 stool isolates from the 2003 AHC epidemic showed a 97.7–100% 3C nucleotide sequence identity and 97.3–100% VP1 nucleotide sequence identity with conjunctival viral isolates (*n* = 23) (Table 1). The 3C and VP1 sequences from the stool isolates were identical to that of the conjunctival viral isolates in four patients with concurrent CVA24v–positive conjunctival and stool specimens. The sequence analysis of the 3C protein-coding regions showed that two stool isolates from the 2008–2009 epidemic had a 99.4–100% nucleotide identity with 14 conjunctival swab isolates from the same epidemic year (Table 1).

### 3.2. Phylogenetics of the 3C and VP1 Gene Sequences of CVA24v Strains Isolated from Feces

There were 15 unique sequences from the CVA24v strains isolated from feces (i.e., 12 3C gene sequences and three VP1 gene sequences) (Appendix A).

Among the 12 partial 3C sequences, eight were obtained from CVA24v strains isolated during the 1997 AHC outbreak, three from the 2003 Cuban AHC epidemic, and the remaining one was from the 2009 Cuban AHC outbreak (Figure 1 and Figure 2, Appendix A). These sequences, identified as genotype IV of CVA24v [13], were compared with partial 3C sequences of CVA24v strains from Cuba (*n* = 42) and 17 other countries (*n* = 83) (Appendix A). All eight fecal CVA24v isolates from the 1997 Cuban outbreak formed a clade together with the 1997 Cuban sequences and 1998 USA sequence (EF015040_Texas_USA_1998) isolated from conjunctival swabs. Two out of three CVA24v strains isolated from feces during the 2003 Cuban AHC epidemic clustered with CVA24v strains isolated from conjunctival swabs in Cuba and Guadaloupe in 2003. The other strain clustered with strains isolated from conjunctival swabs in India (2003) and Brazil (2003–2005) (Figure 1). One CVA24v strain isolated from feces in 2009 was also part of a clade formed by strains obtained from conjunctival swabs during the AHC outbreaks in Cuba (2008–2009), India (2007), and Brazil (2009) (Figure 1).

Among the three partial VP1 sequences included in the phylogenetic analysis, two were obtained from CVA24v strains isolated from feces during the 1997 Cuban AHC outbreak, and the other was obtained from the 2003 Cuban AHC epidemic (Figure 2, Appendix A). All these obtained sequences belonged to genotype IV and were compared with 32 Cuban sequences and 95 partial VP1 sequences from strains isolated in 18 countries (Appendix A). The two fecal CVA24v isolates from the 1997 Cuban outbreak formed a clade together with sequences from CVA24v strains isolated from conjunctival swabs in Cuba (1997) and USA (EF015040_Texas_USA_1998). The remaining VP1 sequences obtained from fecal CVA24v isolates from 2003 clustered with sequences from isolates from conjunctival swabs from Cuba and French Guyana in 2003 (Figure 2). The phylogenetic analysis of CVA24v sequences from AFP cases in the Philippines (*n* = 13), India (*n* = 4), Bangladesh (*n* = 9), and Gabon (*n* = 1), from nonhuman primates (*n* = 3, Bangladesh), and from environmental samples (*n* = 5, Philippines) showed that all CVA24v sequences clustered into three major genotypes within the VP1 region, regardless of the type of sample from which the isolates were obtained (Figure 2).

## 4. Discussion

This study shows that fecal samples are equivalent to conjunctival samples for the genetic characterization of CVA24v, even during AHC outbreaks. Conjunctival swabs are usually considered the ‘gold-standard’ for the diagnosis and molecular characterization of CVA24v-associated AHC outbreaks [32,33]. However, it might be challenging to isolate the virus from conjunctival swabs, particularly if the samples are collected late after the acute phase of the infection. In this study, molecular similarities were identified between CVA24v isolates from fecal and conjunctival swabs from patients with AHC. These findings are of particular importance given that fecal samples are easier to collect and provide a noninvasive assessment of large sample sizes. This makes the identification and isolation of CVA24v possible even after the acute phase has passed, like for other human enteroviruses, which are excreted in feces for a long period of time [32].

CVA24v shedding in the stool of patients with AHC has been well-described since the first AHC epidemic [3,4,5]. CVA24v has also been isolated from non-AHC patients’ feces, in association with AFP cases during AFP surveillance programs in the Philippines, Cameroon, India, and Bangladesh [28,29,30]. At this point, it is important to note the report by Kosrirukvongs et al. [34] on a patient who developed facial paralysis with an incomplete recovery three months after onset during an AHC epidemic in Thailand during 1992. Interestingly, nonvariant CVA24 strains have also been associated with AFP cases in different studies conducted in African countries, Australia, Bangladesh, China, and Bolivia since 1992 [35,36,37,38,39,40,41]. This weight of evidence points to a possible neuroinvasion potential of CVA24v. Thus, heightened attention needs to be paid to the possible contribution of CVA24v to the development of neurological complications.

CVA24v has been isolated from sewage and rivers [31,42], reflecting waterborne and sewage-associated virus transmission routes during epidemics [43]. Moreover, CVA24v sequences have been obtained from the feces of nonhuman primates in Bangladesh and Cameroon, suggesting that nonhuman primates may serve as the temporal reservoir hosts of CVA24v during interepidemics periods [27,41]. It is worth noting that some VP1 sequences of strains obtained from nonhuman primates and environmental samples were included in the dataset of our phylogenetic analysis (Figure 2, Appendix A). Remarkably, sequences from human and nonhuman primate-infecting viral strains formed the same clade, even though the strains were derived from different hosts. These findings point toward host plasticity and a diversity of epidemiological and clinical characteristics among CVA24v strains.

In addition, this study showed that CVA24v strains isolated sequentially over 20 days from the fecal samples of one patient (Appendix A) differed in five and two nucleotide positions in the 3C and VP1 regions, respectively. It is therefore conceivable that the presence of CVA24v in stool stems from the virus’s ability to infect the gastrointestinal cells. Since stool samples from patients with AHC are infrequently sampled by the treating physician, a limitation of our study is the relatively small sample size of fecal samples, particularly consecutive stool specimens from patients with AHC. Therefore, further longitudinal studies are required to follow up a larger number of patients with consecutive stool and conjunctival samples in order to gain a more comprehensive understanding of genetic changes in the context of CVA24v evolution and the persistent release of CVA24v in fecal samples from AHC patients. Whether the viral gastrointestinal infection may contribute to the accumulation of novel genetic variants of CVA24v also needs to be addressed using high-throughput sequencing approaches after direct nucleic acid extraction from the stool.

In conclusion, the findings of this study suggest that fecal specimens can serve as a reliable alternative sample type for the routine molecular diagnosis and molecular epidemiology of CVA24v. Moreover, this study call for further attention to three key features that should be taken into consideration when analyzing the epidemiology of CVA24v: (i) the neurotropism of CVA24v and potential neurological complication related to its high epidemic and pandemic potential, (ii) the environmental presence of CVA24v in sewage rivers and effluents, representing a possible transmission vehicle during epidemics, and (iii) the nonhuman reservoirs as a potential source of CVA24v epidemics. While the role of fecal excretion as a hidden source of the spread of pandemic CVA24v and the magnitude of the health threat posed by CVA24v is grossly underestimated, altogether, this study underscores the importance of feces as an additional route of CVA24v transmission and unveils the diverse spectrum of clinical syndromes caused by CVA24v infection.

## Figures and Tables

**Figure 1 microorganisms-09-00531-f001:**
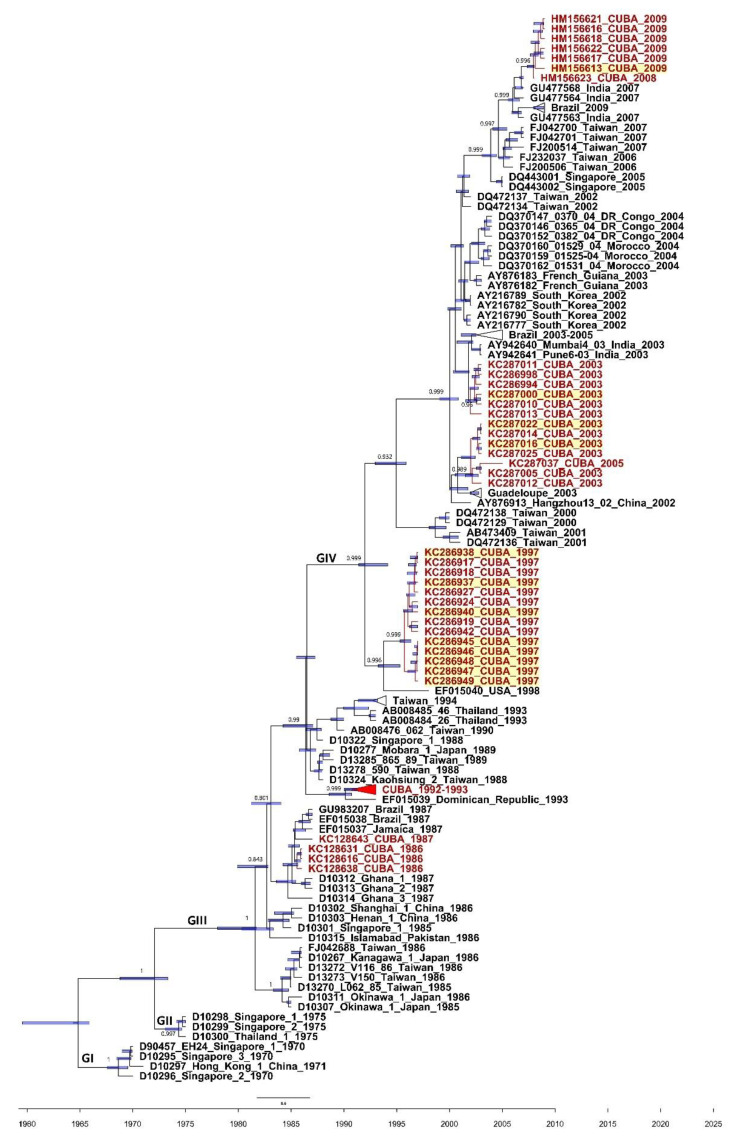
Maximum clade credibility (MCC) phylogeny of the 3C (507 nucleotides) coding region of the 54 Cuban (in red) and 83 worldwide CVA24v strains constructed by Bayesian method with the Bayesian Evolutionary Analysis Sampling Tree (BEAST) v1.8.4 program. Bars at nodes indicate with a 95% highest posterior density (HPD) the time of the most recent common ancestor (TMRCA). Support values with a posterior probability >0.700 are indicated at the nodes. Branches forming genotypes GI-GIV are shown. The sequences are indicated by the GenBank accession number, strain name, country, and year of isolation. The sequences obtained from feces isolates of acute hemorrhagic conjunctivitis (AHC) cases are highlighted in yellow.

**Figure 2 microorganisms-09-00531-f002:**
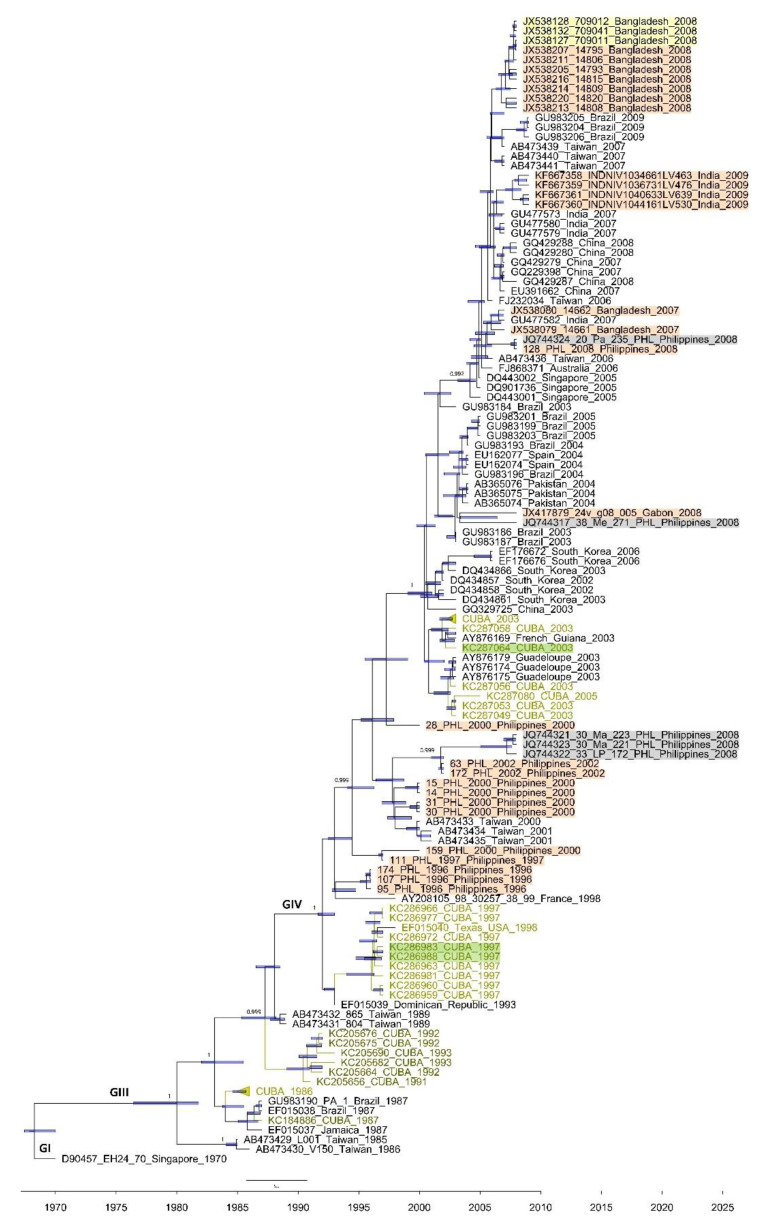
Maximum clade credibility (MCC) phylogeny of the VP1 (234 nucleotides) coding region of 35 Cuban (in green) and 95 worldwide CVA24v strains constructed by Bayesian method with the Bayesian Evolutionary Analysis Sampling Tree (BEAST) v1.8.4 program. Bars at nodes indicate with a 95% highest posterior density (HPD) the time of the most recent common ancestor (TMRCA). Support values with a posterior probability >0.700 are indicated at the nodes. Branches forming genotypes GI-GIV are shown. The sequences are indicated by the GenBank accession number, strain name, country, and year of isolation. Cuban sequences obtained from feces isolates of AHC patients (*n* = 3) are highlighted in green; sequences obtained from Synanthropic nonhuman primate feces isolates (*n* = 3) are highlighted in yellow [27]; the sequences obtained from acute flaccid paralysis (AFP) feces isolates (*n* = 27) are highlighted in orange [27,28,29,30]; and the sequences obtained from Philippines river samples (*n* = 5) are highlighted in grey [31].

**Table 1 microorganisms-09-00531-t001:** Nucleotide identity comparison of 3C and VP1 sequences between stool isolates and those obtained from swabs.

Years	No. of Sequences from Feces		No. of Sequences from Conjunctival Swabs		Nucleotide Identity (%)	
	3C	VP1	3C	VP1	3C	VP1
1997	16	14	25	22	99.0–100	99.1–100
2003	16	16	23	23	97.7–100	97.3–100
2008–2009	2	2	14	-	99.4–100	-

## Data Availability

All data supporting the findings of this study are included within the article and its supplementary information file. A list of NCBI accession number of all sequences analyzed is available in Appendix A.

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
