# Peer review of "Molecular Characterization of Coxsackievirus A24v from Feces and Conjunctiva Reveals Epidemiological Links"

_microorganisms, 2021, doi:10.3390/microorganisms9030531_

Round 1
Reviewer 1 Report
Fonseca et al present a very interesting piece of work on the CVA24 variant from conjunctiva and stool, in the context of other human and non-human isolates. They demonstrate the utility of stool samples as an alternative sample source in outbreaks of acute haemorrhagic conjunctivitis - this is relevant for diagnostics and epidemiology as the shedding in stool is considerably longer and of higher quantity as compared to that from the eye. A nice piece of work, with a clear research question, very thorough analysis of the phylogenetic relatedness, and a comprehensible style.
I have only a handful of comments.
(a) The methods are adequate albeit a bit old-fashioned. They indeed may have benefited from a revamp. Namely, although the viruses were successfully isolated on cell culture, some of them did not yield sequencing signal. This is hardly acceptable nowadays - the authors should have designed additional primers, or resort to the use of metagenomic sequencing the culture supernatant. This weakens the main finding of the paper - the samples with no sequence most likely contained other considerably distant sequences as compared to the rest of the collection, otherwise they would amplify.
(b) The replacement of nucleotides in viral genome occurs gradually over time. Therefore it would be more suitable to investigate the subset of repeated stool samples directly from the material, in order to observe the evolving quasispecies. Also here the classic Sanger sequencing may be a bit insensitive. Its use however does not invalidate the observation of long-term carriage of the same strain, the authors provide a convincing proof!
(c) Very minor issues: 1. Table 1 uses slashes for some reason - why, there is no proportion presented. 2. Lines 161-169 and onwards kindly double check nomenclature (strain, isolate, clade).
Reviewer 2 Report
Molecular characterization of Coxsackievirus A24v from feces and conjunctiva reveals epidemiological links.
In this study the authors compared the 24 VP1 and 3C sequences of CVA24v strains isolated from feces and conjunctival swabs during acute hemorrhagic conjunctivitis epidemics in Cuba 25 in 1997, 2003, and 2008-2009.
There is a high degree of nucleotide identity (ranging from 97.3-100%) between stool samples and the corresponding conjunctival isolates. The molecular features of CVA24v in feces have not yet been studied.
Methods:
This study actually describes the 24 VP1 and 3C sequences of CVA24v strains comparisons isolated from feces and conjunctival swabs between 1997 and 2009 as before 1997 there were no fecal samples available. From the 1997 sampling: “Two isolates were from one patient, from whom CVA24v also be isolated in nasal and pharyngeal swabs, and seven isolates derived from one patient with sequential fecal sampling.”
- I would like to suggest to the authors to select ONE sample (fecal and conjunctival swabs) per patient for further comparison as it is very unlikely that the sequences within one patient will alter dramatically, unless there is some serious underlying condition within one patient that facilitates chronic AHC and accompanying conjunctiva and fecal sampling.
- The authors actually only compare paired samples from the years: 1997, 2003, 2008 and 2009. The years in between were either not collected, or there were no epidemics or there were no paired samples (fecal and conjunctival swabs) available. This should be changed throughout the manuscript.
- Section 2.2: “Viral RNAs from the 1986-2003 CVA24v strains..” should be “Viral RNAs from the 1997-2003 CVA24v strains…” How were the viral RNAs extracted from the 2003-2009 CVA24v strains?
Discussion:
- The authors state: Moreover, the virus is excreted for a longer period in feces than the conjunctiva. It would be really nice if the authors would highlight the VP1 and 3C sequence of the one patient in a graph that was sampled for 6 times and was shedding for more than 20 days as a proof of concept to show indeed that fecal shedding is relevant for diagnosing AHC.
The conclusion is good and appropriate.
